# Designing a Library of Lived Experience for Mental Health: integrated realist synthesis and experience-based co-design study in UK mental health services

Paul Marshall [1] , John Barbrook,[2] Grace Collins,[3] Sheena Foster,[1] Zoe Glossop [1] , Clare Inkster,[4] Paul Jebb,[5] Rose Johnston,[1] Steven H Jones,[1] Hameed Khan,[1] Christopher Lodge,[1] Karen Machin,[6] Erin Michalak,[7] Sarah Powell [8] , Samantha Russell,[1] Jo Rycroft-Malone,[9] Mike Slade,[10,11] Lesley Whittaker,[5] Fiona Lobban[1]

**Correspondence to**
Dr Paul Marshall;
p.marshall4@lancaster.ac.uk

## ABSTRACT

**Objective** Living Library events involve people being trained as living 'Books', who then discuss aspects of their personal experiences in direct conversation with attendees, referred to as 'Readers'. This study sought to generate a realist programme theory and a theory-informed implementation guide for a Library of Lived Experience for Mental Health (LoLEM).

**Design** Integrated realist synthesis and experience-based co-design.

**Setting** Ten online workshops with participants based in the North of England.

**Participants** Thirty-one participants with a combination of personal experience of using mental health services, caring for someone with mental health difficulties and/or working in mental health support roles.

**Results** Database searches identified 30 published and grey literature evidence sources which were integrated with data from 10 online co-design workshops conducted over 12 months. The analysis generated a programme theory comprising five context-mechanism-outcome (CMO) configurations. Findings highlight how establishing psychological safety is foundational to productive Living Library events (CMO 1). For Readers, direct conversations humanise others' experiences (CMO 2) and provide the opportunity to flexibly explore new ways of living (CMO 3). Through participation in a Living Library, Books may experience personal empowerment (CMO 4), while the process of self-authoring and co-editing their story (CMO 5) can contribute to personal development. This programme theory informed the co-design of an implementation guide highlighting the importance of tailoring event design and participant support to the contexts in which LoLEM events are held.

**Conclusions** The LoLEM has appeal across stakeholder groups and can be applied flexibly in a range of mental health-related settings. Implementation and evaluation are required to better understand the positive and negative impacts on Books and Readers.

**Trial registration number** PROSPERO CRD42022312789.

## STRENGTHS AND LIMITATIONS OF THIS STUDY

⇒ This study used a novel, iterative and creative approach to integrating theory development and intervention co-design.
⇒ A key strength of this approach was the involvement of people with lived experience expertise in mental health at every stage of co-design and theory development.
⇒ The programme theory and implementation guidance were informed by analysis of research on previous Living Libraries and detailed co-design workshops, which drew on broad professional and personal mental health experiences.
⇒ However, few evidence sources identified by systematic searches describe Living Libraries focused specifically on mental health.

## INTRODUCTION

The value of sharing health-related experiences is widely recognised. Varied contexts draw on these experiences, including for shaping research,[1] enriching professional education[2] and informing peer support.[3] There is an expanding evidence-base focused specifically on mental health lived experiences. For example, social contact interventions reduce mental health stigma,[4] mental health peer support contributes to improvements in psychosocial outcomes[5] and there is growing interest in the use of mental health narratives to achieve a range of organisational aims.[6] The importance of integrating lived experience perspectives in health service development is acknowledged in UK and global policy,[7 8] emphasising the need for continued efforts to promote lived experience perspectives.

Existing models for sharing mental health experiences do have limitations. Many

opportunities are offered only periodically by health services and educational institutions, such as those linked to service improvement[9] or health professional training.[10] Others require commitment to more formal positions, including the peer-support worker role in the UK,[11] which may preclude access by those with existing commitments. There is an outstanding need to extend such opportunities to groups whose voices are often not heard in mainstream conversations about mental health, including those from marginalised communities.[12]

Adaptable approaches are required for expanding opportunities to share and learn from mental health experiences. The Living Library may represent one such model. Pioneered by the Human Library Organisation,[13] these events involve people being supported to share aspects of their life experience in direct conversations with others. Those sharing their stories are called living 'Books', while those listening are termed 'Readers'. Conversations often last around 20 min and involve a Book briefly describing their personal narrative, then being open to questions from the Reader.[14] Events are facilitated by staff, or 'Librarians', who provide guidance to those involved. These events are often held in open settings, including public and university libraries and have typically aimed to challenge a range of societal prejudices by facilitating interactions between people who may not otherwise engage in conversation around topics such as personal experience of racial prejudice, experience of ill-health and gender-based discrimination.[15] The model has also been used in a mental health context to address stigma[16] and facilitate peer-support.[17] However, implementation recommendations sensitive to the specific challenges of discussing mental health experiences are limited.

Following UK complex intervention development guidance calling for greater emphasis on theory development and stakeholder engagement,[18] we applied a novel integration of realist synthesis and experience-based co-design (EBCD) to explore the Living Library as a strategy for sharing mental health experiences. Realist synthesis is a method of evidence synthesis used to develop causal statements, or programme theories, to explain how social programmes work, for whom and under which circumstances.[19] EBCD identifies opportunities for healthcare innovation and draws on stakeholder perspectives to generate creative solutions.[20] Our integrative approach is detailed in a published research protocol.[21]

### Objective

▶ To use realist synthesis to develop a programme theory for a mental health-focused Living Library, which we term a Library of Lived Experience for Mental Health (LoLEM).
▶ To use insights from this synthesis to inform co-design workshops with a range of mental health stakeholders, with the goal of developing an accessible implementation guide.

## METHODS

This study received ethical approval from Coventry and Warwick National Health Service Research Ethics Committee (ref: 305975). The systematic search strategy was developed with an information specialist (JB) and preregistered on PROSPERO https://www.crd.york.ac.uk/prospero/display_record.php?RecordID=312789. RAMESES guidance informed the reporting of this study.[22] A Consolidated criteria for Reporting Qualitative research checklist is presented in online supplemental file 1.

As described in the study protocol,[21] theory development and EBCD workstreams were iterative and ran in parallel. However, the study broadly progressed through the following stages which represent an integration of the steps taken when conducting realist synthesis and EBCD.[19 20]

### Eliciting initial theories and touchpoints

Initial programme theories (IPTs) were elicited through theory gleaning interviews[23] with six members of an expert advisory group and the study public and patient involvement (PPI) lead who were invited to contribute to the study based on their experience participating in or researching Living Libraries in the domains of mental health (EM, MS, SF, CL), education (SP) and stigma reduction (CI, HK) (topic guide available in online supplemental file 2). Interviews were conducted and recorded using the online video software Microsoft Teams by PM, a male research associate with doctoral level experience in qualitative interviewing. PM had no previous experience of Living Libraries prior to conducting the interviews. As members of the research team, all interviewees had prior relationships with the interviewer and were fully informed of the reason for their participation. Twenty IPTs were developed based on initial coding by PM, FL and RJ. Researcher bias was mitigated through written feedback on initial drafts and advisory group discussion prior to the IPTs being finalised (online supplemental file 3). IPTs highlighted key issues to be explored with participants in co-design workshops, referred to as 'touchpoints' in the EBCD process.[20]

### Retrieving relevant evidence

Searches were conducted on research databases and grey literature sources (figure 1) from 2000, the inception of the initial Human Library approach,[13] to March 2023. Preparatory unsystematic searches indicated that relatively few articles in this literature were likely to focus solely on mental health. Therefore, reports on Living Libraries related to specific topics including mental health and from generalist events featuring a range of lived experiences were included. Following title and abstract screening, the full-texts of identified articles were judged for inclusion using a realist-informed assessment of relevance and rigour (PM) with 20% independently assessed by a second reviewer (RJ). Citation chaining was used to identify articles in reference sections and

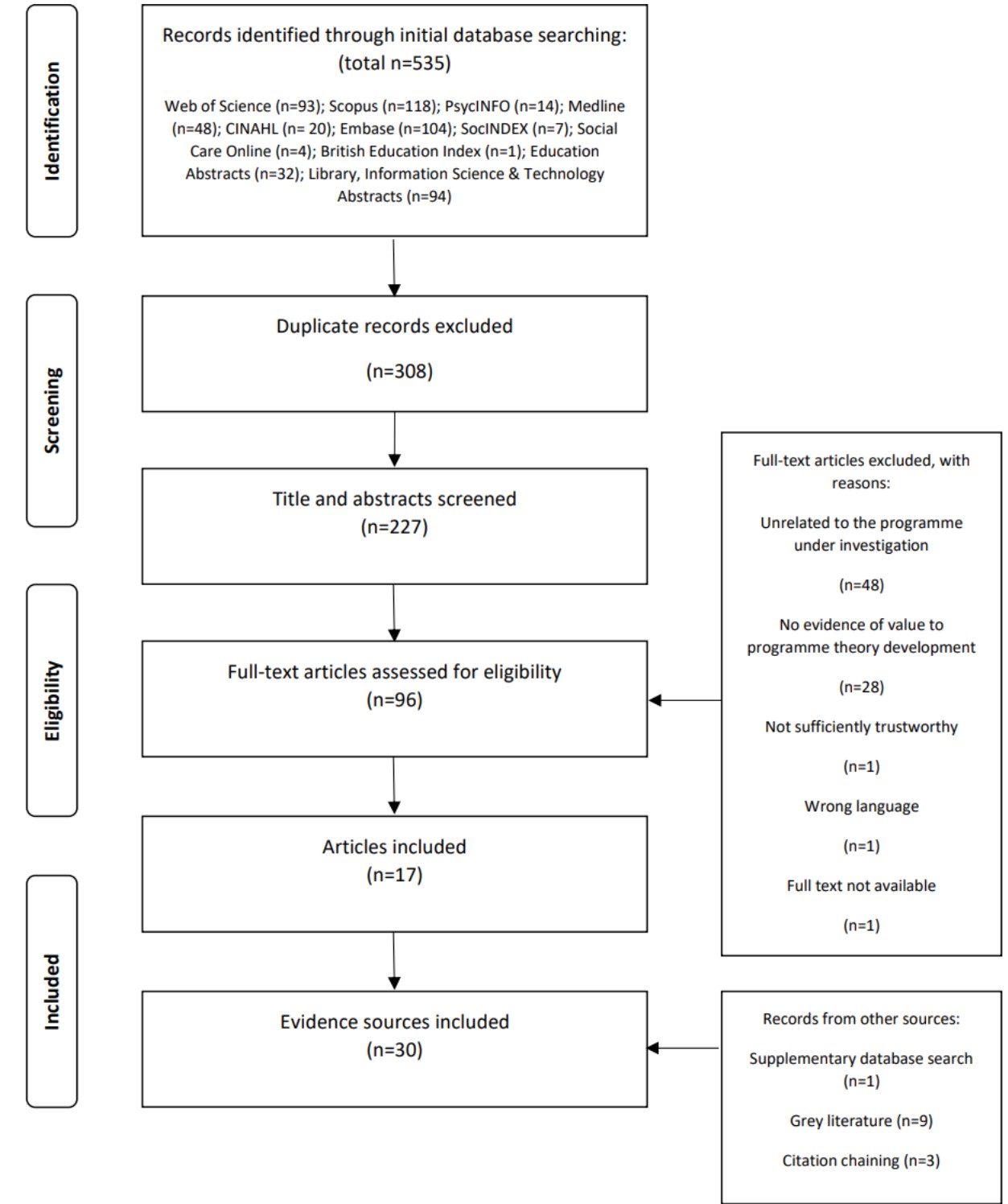

**Figure 1** Flow diagram of evidence identification.

'cited by' pages of articles included in the initial search. IPTs and initial workshops highlighted psychological safety as a key theoretical and implementation focus for this study. Consistent with the iterative nature of realist synthesis searches,[24] we conducted a targeted search of a large multidisciplinary database for existing formal theories related to the concept and included an additional evidence source.[25] A full description of each search is available in online supplemental file 4.

### Iterating initial theories
Following guidance on the use of qualitative analysis software for realist analysis,[26] IPTs were added to NVivo V.12 as nodes to create an initial programme theory

framework. PM conducted coding of the full text of each evidence source. This involved identifying and labelling sections of text related to potential contextual factors, mechanisms and/or outcomes, as defined in realist methodology.[19] Codes were linked to the most relevant initial programme theory node. PM, FL and RJ refined the initial theory framework by reformulating, consolidating and combining initial statements based on whether underlying data supported, refuted or refined individual IPTs. While reviewing this evidence, retroductive reasoning was used to hypothesise causal forces behind regularities apparent in the data. An iterative process of written feedback and group discussion led to the consolidation of the initial framework into a series of candidate context-mechanism-outcome (CMO) configurations.

### Establishing a multistakeholder co-design group

Eligible EBCD participants were adults with any self-identified mental health experience. We recruited participants by distributing digital study advertisements to local mental health services, health research networks and third sector mental health support organisations primarily based in the North of England. Workshops were facilitated by KM and GC, both experts in mental health peer-support who deliver workshops from a lived experience perspective. Workshops were supported by researchers (ZG, RJ, PM), senior nurses (PJ, LW), a service user researcher (CL) and academic clinical psychologist (SHJ).

### Designing workshops reflexively

We ran 10, 2-hour online co-design workshops over 12 months using the online video software Zoom. Workshops were flexibly designed in team meetings around two primary goals. First, we aimed to develop a comprehensive implementation guide. This required the team to reflect on gaps in knowledge and plan further exploration of topics pertinent to implementation. Second, we explored 'touchpoints' to facilitate programme theory refinement. One such example was the exploration of psychological safety in workshop 4, a key concept in the IPT framework which required further investigation to understand relevant implementation factors.

### Developing and refining outputs

Workshop participants' views and implementation suggestions were captured using the collaborative online note taking platform Jamboard [27] and in researcher (CL, PM, RJ, SHJ, ZG) field notes. Data were combined in workshop summary documents following each session, reviewed by the research team in post-workshop debriefs and added to NVivo for integration into the ongoing analysis.

Summary documents informed an initial draft implementation guide, refined through feedback from the EBCD group. Implementation recommendations from the co-design process were linked to the developing theoretical framework. CMOs were further refined through discussion and written feedback from the wider research team and expert advisory group.

Final versions of the LoLEM programme theory and implementation guide were shared during an interactive dissemination event at which the LoLEM was piloted and the guide made freely available. [28]

### Public and patient involvement

Experts by experience in mental health contributed to all stages of this study. A stakeholder group informed the initial study design and experts by experience were co-applicants for funding. The study team included a service user research and experts by experience were involved in an advisory group, all of whom are involved in authoring key study outputs. The co-design group involved service users and carers and was led by facilitators who delivered workshops from a lived experience perspective. Participants and team members with lived experience assisted with dissemination at local and national events.

## RESULTS
### Results of systematic searches

Database and grey literature searches returned 30 eligible evidence sources. Figure 1 describes the process of evidence identification. Study characteristics are available in online supplemental file 5.

### Description of co-design workshops

EBCD workshops were attended by 31 participants, 16 of whom identified as women and 15 men. Four had previously been involved in hosting a Living Library event and one had experienced being a living Book. Workshops were attended by people with experience of using mental health services (n=23), informal caregiving (n=13) and working in mental health services (n=8) or in voluntary sector mental health support roles (n=8). Participant demographic details are available in online supplemental file 6.

Details of activities completed, and mental health experiences represented at each workshop, are presented in table 1.

### LoLEM programme theory

We developed a programme theory comprising five CMO configurations, with corresponding implementation recommendations (figure 2).

#### CMO 1—psychological safety facilitates dialogue

When event organisers implement guidance and support strategies tailored for their organisational settings (context), participants will be better able to understand the Living Library model and how to manage personal boundaries such that they feel in control of their experience (mechanism). This will promote participants' perceptions of psychological safety (outcome) necessary for engaging and productive conversations.

Psychological safety refers to an individual's comfort with taking interpersonal risks to meet their goals within a

**Table 1** Experience-based co-design workshop details

| Workshop number | Total participants | Experience using mental health services | Experience as informal carers | Experience as mental health service staff | Experience in voluntary mental health support roles | Workshop activities |
|---|---|---|---|---|---|---|
| 1 | 25 | 19 | 8 | 4 | 7 | ► Project introduction.<br>► Story formulation activity. |
| 2 | 27 | 18 | 11 | 5 | 7 | ► Story sharing activity.<br>► Reflection on personal boundaries. |
| 3 | 25 | 21 | 10 | 4 | 8 | ► Creating a fictional Reader 'persona'.<br>► Reflection on Library design for different Reader groups. |
| 4 | 24 | 20 | 10 | 2 | 8 | ► What influences potential Readers' decisions to participate?<br>► What constitutes a safe space for sharing mental health experiences?<br>► How can event organisers cultivate psychological safety among Books and Readers? |
| 5 | 25 | 19 | 11 | 2 | 8 | ► Reflection on participants' preferred roles at a Living Library (Books, Readers or Librarians).<br>► Planning a hypothetical Living Library.<br>► A 'design to fail' activity, considering factors likely to contribute to unsuccessful events. |
| 6 | 23 | 17 | 8 | 3 | 6 | ► Design considerations for Living Libraries in different contexts (the voluntary, the National Health Service and public libraries). |
| 7 | 23 | 18 | 9 | 2 | 7 | ► Identifying resources required to run successful Living Libraries.<br>► A 'sales pitch' activity to potential funders. |
| 8 | 21 | 16 | 9 | 2 | 5 | ► Advantages and disadvantages of delivering Living Libraries with different aims (mental health support vs changing public perception of mental health).<br>► Co-design of the implementation guide contents page. |
| 9 | 20 | 15 | 10 | 2 | 5 | ► The desired characteristics of event staff ('Librarians').<br>► Considering how to evaluate a Living Library.<br>► Feedback on the implementation guide. |

**Table 1** Continued

| Workshop number | Total participants | Experience using mental health services | Experience as informal carers | Experience as mental health service staff | Experience in voluntary mental health support roles | Workshop activities |
|---|---|---|---|---|---|---|
| 10 | 17 | 15 | 7 | 2 | 3 | ► Celebration event. <br> ► Review of project successes and thanking participants for their contributions. |

given organisational context.[25] The Living Library model provides a platform for participants to take interpersonal risks by sharing experiences in direct conversation. We propose that psychological safety creates a facilitating context, or 'ripple' effect,[29] promoting the likelihood of subsequent positive impacts described by CMOs 2–5.

Developing psychological safety for Books and Readers commonly involves implementing event rules to explicitly promote mutual respect, awareness of boundaries and a sense of personal control. Pre-event briefings may be used to reinforce the principles of the approach: 'Readers are required to return the 'Book' in the same psychological and physical condition and are asked: 'Never harm a Book!'.[30] Promoting psychological safety for Books can involve offering training and practice sessions to set

expectations, identify what they wish to share and prepare for questions they may not want to answer. EBCD participants suggested that psychological safety would further be promoted by the reassuring presence of attentive staff, or 'Librarians', able to provide guidance, oversight and emotional support where necessary.

Given their conversational focus, Living Libraries hold potential for Books and Readers to experience interpersonal challenges that could undermine psychological safety. Books may be exposed to negative attitudes towards mental health, feel pressure to be an exhaustive authority on the issues they discuss, 'pumped' for knowledge by repeated readings, or alternatively, 'left on the shelf' by uninterested Readers.[31 32] Readers themselves may experience discomfort hearing directly about others'

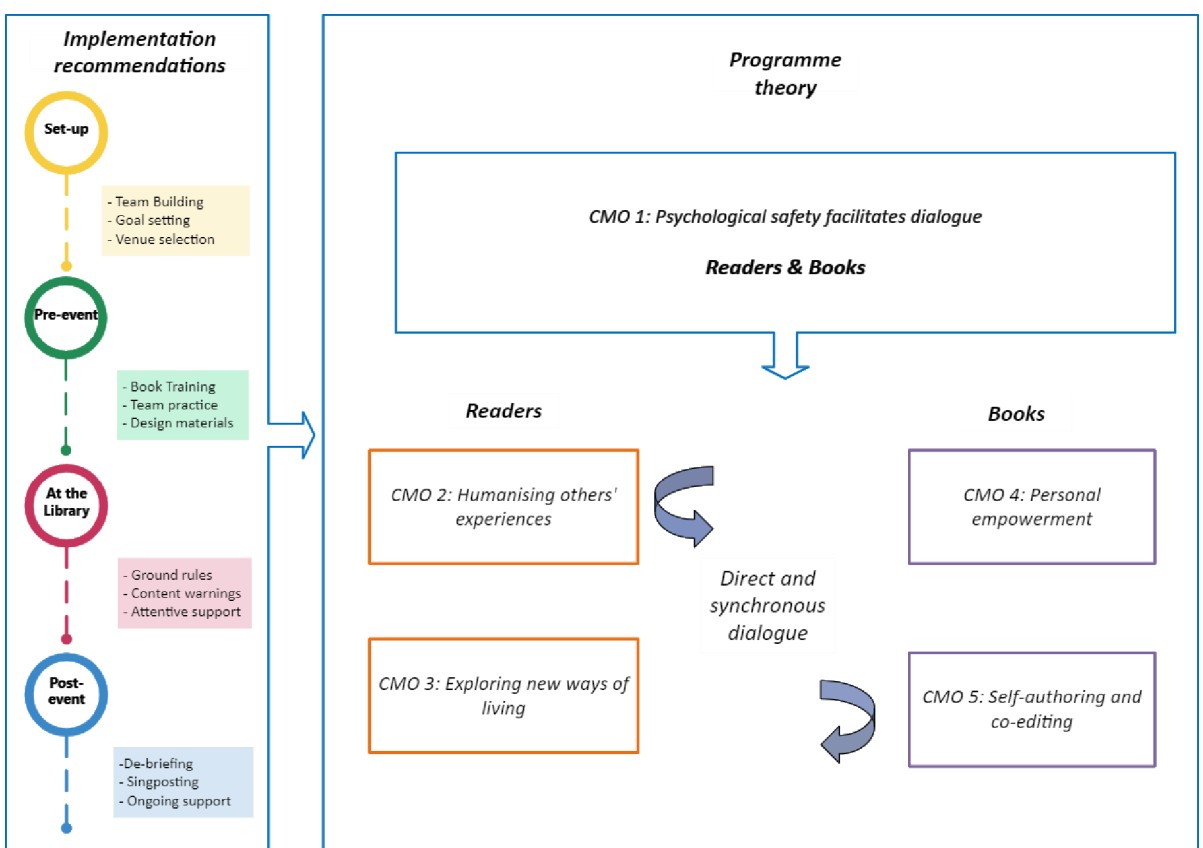

**Figure 2** Library of Lived Experience for Mental Health programme theory with implementation recommendations. CMO, context-mechanism-outcome.

**Table 2** Programme theories with illustrative data extracts

| Programme theory | Illustrative data | Source |
|---|---|---|
| Psychological safety facilitates dialogue | '[Books'] needs must be recognised and met. This could include things like limiting the number of readings a living book can be required to participate in within a given session, providing adequate support for living books including those with special needs, and providing time and structure for living books to debrief after readings or to read other living books.' | [42] |
| | ' [a living library conversation] will be a deep dive, might need to help these first-time Books prepare for example, with practice and storytelling and thinking about boundaries' | Workshop 2 Jamboard |
| Humanising others' experiences | 'Readers were given permission to empathize with the experiences of Books and, in the process, see themselves in the Books' stories.' | [35] |
| | 'Stories trigger a response in us that helps us empathise with others.' | Workshop 3 field notes |
| Exploring new ways of living | 'Another Reader found inspiration in a Book's journey that he could apply to his own life: "I learned from some people's experiences that, okay, I'm going through this right now, but you went through it and you overcame it. So I have a choice".' | [43] |
| | '(Books are] providing other visions/real life examples of ways forward that are hopeful.' | Workshop 8 Jamboard |
| Personal empowerment | 'Living libraries excel as a strategy for giving voice to marginalised groups. Living library conversations allow for direct self-representation, unmediated by third parties.' | [41] |
| | 'Mental health has been stigmatised for so long, people ignored, locked away and forgotten. Just the very action of telling someone that you want to hear their story is massive.' | Workshop 2 Jamboard |
| Self-authoring and co-editing | 'By thinking about and structuring their story, people will often 're-author' their lives, by defining their own existence in relationship to themselves and what they were going through at the time; thereby constructing reality by choices made to give meaning to their lives.' | [54] |
| | 'Storytelling can be therapeutic for teller.' | Workshop 2 Jamboard |

distress. The emergent nature of these interactions means that such circumstances cannot be entirely predicted, yet event planners should prepare for and mitigate these eventualities as far as possible by foregrounding implementation strategies to promote psychological safety.

### CMO 2—humanising others' experiences

Direct conversations with Books (context) humanise the experience of mental health difficulties (mechanism), contributing to Readers developing a greater empathetic understanding of mental health (outcome).

Event organisers framed direct interaction between Books and Readers as an explicit goal. Many aimed 'to bring people together who may not otherwise meet and to also kind of shut down the stereotypes that people have about others'.[33] Events focused on challenging prejudice sought to provide a platform for Books to 'show the general public they are 'human''[31] and challenge superficial views of important aspects of their experiences and identities by breaking down perceived 'us-them' divides between Books and Readers.[34] This could occur through recognition of shared experience including emotional difficulties, to which Readers could relate regardless of personal differences: 'Even women [who don't] look like me, or identify as the same race as me still go through

similar struggles. So that was eye-opening because I've never really thought about it like that'.[35]

This more nuanced understanding of others' experiences was for some facilitated by the subtle emotional and behavioural cues present in direct interaction, which augmented the interpersonal connection shared by Books and Readers.[33 36] In this context, the experiences and identities Books represent shift from abstract and disembodied concepts to personified and meaningful human experiences.[37 38] Mental health difficulties are thus reconceptualised from 'myths to storied realities',[39] enhancing Readers' abilities to subsequently understand and empathise with the perspectives of people experiencing similar difficulties.[34]

### CMO 3—exploring new ways of living

When Living Libraries provide Readers with the permission to explore Books' experiences through synchronous interaction (context), they will use this opportunity to flexibly explore issues of personal significance (mechanism). This will facilitate awareness of new and helpful ways of living (outcomes).

The synchronous nature of Living Library conversations allows Readers to personalise their interactions by asking questions about aspects of Books' stories that

resonate with their own. As noted by a participant in a mental health-specific event,[34] 'in a talk, listeners listen passively. As speakers, we are asked to tell our whole story. But in the Human Library, it does not matter whether the sharing is complete or not. The important thing is to let readers know what they want to know'. This form of experiential learning can influence how Readers understand ways of managing their own distress[17] and may help health professionals identify new ways of supporting service users.[38 40]

EBCD participants noted that the relative novelty of the Living Library approach and the associated 'Book' metaphor can imply that the sharing of experience is intended to be unidirectional, with Readers taking a passive role. Organisers can facilitate interactive dialogue by providing Readers with explicit permission to explore questions of personal significance, within established boundaries. Strategies for reaffirming Readers' conversational permission include providing library attendees with clear ground rules and example questions or cues to prompt engagement.

### CMO 4—personal empowerment

When Living Libraries facilitate the authentic expression of Books' personal experiences (context), Books will feel that their expertise has been heard and valued (mechanism), contributing to personal empowerment (outcome).

Living Libraries have been used to spotlight marginalised voices.[41] The approach both promotes Books' sense of having been heard as experts by experience and provides a medium to use their stories to meet personally valued goals, including shaping public attitudes and offering support to others. Personal empowerment therefore emerges from this opportunity for direct self-representation, recognition and the pursuit of positive change.[42 43] EBCD group members highlighted the potential of a LoLEM to meet the motivation of many experts by experience in mental health to use their unique perspectives to inspire individual, organisational and social progress against a wider context of stigma and under-recognition. This was mirrored in literature describing participants' sense of pride and satisfaction after sharing their stories at Living Library events.[38 39 44]

The EBCD group also identified how personal empowerment can occur through meaningful participation in event design and delivery. Extensive involvement of experts by experience was suggested to diminish power imbalances between those with lived experience and organisations, such as health services and universities, that may host a LoLEM. Workshops highlighted that the greater the degree of lived experience involvement and collaborative working alongside staff, the more likely a LoLEM is to reflect the perspective of the groups it seeks to engage. It was suggested that this may reduce the potential for disempowerment emerging from peoples' stories being misused.

### CMO 5—self authoring and co-editing

When Living Libraries support Books with developing a personal narrative to share in conversation with Readers (context), Books will explore and develop insight into these aspects of their lives (mechanism). This leads Books to develop new ways of understanding and sharing their experiences which can contribute to personal growth (outcome).

For Books, authoring and articulating a personal narrative represents a 'self-directed process of discovery'[42] and ongoing engagement with story sharing can 'demonstrate how their personal identities evolve and develop over time'.[44] This process may contribute to the reframing of mental health difficulties as an aspect of past experience that is of value to the present self.[39] Relatively unstructured dialogue may also facilitate a form of narrative co-editing: '…in the process of creating a narrative in cooperation with readers, books actually alter their understanding of their own self-appointed topic and what it means to them'.[42] Personal growth can therefore result from a shift in Books' perceptions of their current circumstances through reflexive story refinement: 'Despite occasional moments of discomfort and, perhaps in some cases, because of them, Books recognized that their stories changed because of their participation in the HL [Human Library] Project. Details were added, elements they believed were less important emerged as such, and overall, they achieved greater clarity about their narratives'.[35]

Table 2 provides illustrative data supporting each CMO.

### Implementation guidance

The integrated realist synthesis and EBCD process contributed to the development of a theory informed implementation guide. Recommendations are summarised here with reference to key stages of event delivery (table 3). The full guide is available online.[28]

### DISCUSSION

This integrated realist synthesis and EBCD study generated a programme theory for a LoLEM and theory-informed implementation recommendations. Results emphasise the importance of psychological safety for facilitating productive conversations between Books and Readers. Implementation recommendations highlight ways in which potential organisers may seek to foster supported environments for sharing mental health experiences and further suggestions are reported in a co-designed implementation guide.[28]

This study builds on implementation recommendations for the Human Library Organisation's generalist approach.[14] A notable difference in the context of mental health is the centrality of psychological safety, defined as the ability to take desired interpersonal risks. Occupational research indicates that psychological safety is associated with proactive communication including concern-raising within teams, individual and team-level

**Table 3** Co-designed implementation recommendations by event delivery stages

| | |
|---|---|
| *Set-up* | |
| Team building (CMO 1, CMO 4) | Ensure the event team includes people with the skills to plan, promote and implement a mental health-related event. Embed the perspectives of experts by experience, including in the mechanism by which stories are selected. Recruit books with diverse relevant experiences. Where applicable, provide context-appropriate and timely payment. |
| Goal setting (CMO 1) | Identify a clear goal for the event. Align this with plans for recruitment, support provision and promotion. |
| Venue selection (CMO 1) | Select a non-stigmatising venue that provides a degree of privacy for individual discussions yet does not leave participants feeling isolated and exposed in conversations with people they may not have met before. Conduct accessibility and safeguarding assessments. |
| *Pre-event* | |
| Book training (CMO 5) | Provide comprehensive training to Books. Focus on helping Books to understand the model, what is expected of them and where to go for support. Clearly identify personal boundaries for story sharing and how to manage them. Generate a synopses and story title. Consider developing individualised support plans. |
| Team practice (CMO 1, CMO 5) | Rehearse how the Living Library will be delivered. Books may wish to practice sharing their stories with team members and/or each other to refine their story and prepare for the event. |
| Develop materials (CMO 4) | Develop event materials alongside experts by experience. This may include guidance for readers highlighting what the event is for, rules for participation, and support materials. |
| *At the Library* | |
| Ground-rules (CMO 1, CMO 2, CMO 3) | Clearly outline ground-rules for conversations. This may include instructions for engaging in respectful dialogue and for Books and Readers to be mindful of any boundaries for their conversations, such as to avoid certain topics. Encourage readers to explore topics and questions of personal interest within these boundaries. |
| Content warnings (CMO 1) | If appropriate, consider generating content warnings which could be included in Books' synopses. |
| Attentive support (CMO 1) | Ensure that emotional support is consistent with the nature and scale of the event. Develop plans for managing distress, including risk-related scenarios. |
| *Post-event* | |
| Debriefing (CMO 1) | Implement individual or group debriefs. Consider providing Readers with the opportunity to reflect on their experiences, for example, in a quiet space away from the main event. |
| Signposting (CMO 1) | If appropriate, make available relevant and up-to-date sign-posting material. |
| Supervision (CMO 1, CMO 5) | Consider ongoing supervision for Books and Librarians. This could include a review of any difficulties, support needs or changes to Books' stories. Where available, discuss further opportunities for lived-experience involvement within the organisation. |

CMO, context-mechanism-outcome.

learning and work engagement.[25] Recommendations for promoting psychological safety within this literature suggest organisations should seek to create cultures defined by collaboration and interpersonal openness.[45] This aligns with established principles-based approaches to mental health peer-support,[46] which point to the significance of safe and trusting relationships within organisations that draw on lived experience and control over how those experiences are shared. Results here suggest that by investing in training and building meaningful relationships with those sharing their stories, LoLEM organisers can clearly articulate expectations and promote informed personal disclosure. Drawing on established measures,[25] further research could evaluate the extent to which these practices influence Books' perceptions of psychological safety.

Values underpinning mental health peer support, such as mutuality and reciprocity,[47] are reflected in evidence of positive impacts among those delivering and accessing these services. For example, benefits for those receiving support include improvements in social functioning, hope and stigma reduction, while peer supporters may experience greater confidence, skills development and positive reframing of personal identity.[3] Building on this evidence, results of this study suggest that a LoLEM could provide opportunities to learn about ways of understanding and managing mental health from people who may be empowered by being supported to share their stories. Indeed, participants in a Living Library focused on bipolar self-management welcomed the opportunity for direct contact with peers,[17] demonstrating the feasibility of using this approach in the peer support context. A

LoLEM could also assist service users with sharing stories as part of a recovery-oriented programme of support.[34] Indeed, there is an established body of literature describing the use of storytelling in mental health[6 48] indicating that developing a personal narrative can be highly meaningful for the teller, yet also potentially emotionally challenging.[49] This reinforces this study's emphasis on dedicated support with story formulation for potential Books.

The LoLEM approach requires adaptation for specific contexts. Organisers should consider factors that could undermine the safe and empowering implementation of lived-experience focused programmes. Survivor research identifies how power imbalance between those with lived experience and health services could serve to discourage perspectives deemed unacceptable to institutional agendas, including critical and marginalised voices, or delegitimise those who draw on their experience in pursuit of systemic change.[50] A recent systematic review of lived experience narratives drew attention to potential misuses of mental health experiences, including commodification, coerced elicitation, harm to narrators or audiences and curatorial decisions that limit diversity.[6] Good practice recommendations relevant to the LoLEM model include proactive reflection on organisers' personal biases to ensure power imbalances are not reinforced, encouraging audiences to endorse ethical listening and openness to difference, seeking consent and promoting control over personal narratives.[6] Consistent with this, findings here emphasise the vital role of lived experience perspectives in implementing transparent and collaborative LoLEMs. Established models for involvement, including EBCD, could be used to facilitate development of future programmes.

## Implications for research

Further work may explore the feasibility of Living Libraries as sustained programmes and whether effects persist over time. CMOs reported here could inform the selection of outcomes. The humanising impact of a LoLEM would be expected to contribute to more empathetic perspectives on mental health and in the context of peer-support, future research may investigate potential impacts on Reader self-efficacy. For Books, the proposed positive impact of storytelling and personal empowerment aligns with definitions of personal recovery for which several assessments have been developed.[51] Factors underpinning differences in outcomes across varied service settings, and how this model may fit alongside existing lived experienced-based interventions, are currently unknown. These issues could be addressed using realist evaluation methodology.

## Implications for practice

While lived experience opportunities continue to grow within health services, most roles do not involve training in self-disclosure.[52] The LoLEM model could bridge the gap between people with mental health experiences seeking opportunities to help others, and more formal paid positions requiring considerable time commitments. The approach could also complement efforts to increase lived experience expertise in health professional training, as evidenced by its successful implementation in social work training.[38] More specifically, a LoLEM could afford healthcare trainees opportunities to engage with a broader range of people than they may typically meet, and in neutral settings outside of clinical environments which may be more conducive to open conversation. It is also feasible to combine a Living Library with structured reflexive activities to embed learning and consider the applicability of a Book's story to a Reader's clinical practice.[40] Actions to promote equality, inclusion and diversity in the UK health service should apply an intersectional lens.[53] It has been noted that the Living Library model is well suited to assist health professional trainees with developing a lived appreciation of intersectionality through conversations that illuminate how aspects of identity interact.[35] As per the programme theory reported here a LoLEM could serve to humanise and promote the understanding of nuanced relationships between mental health experiences and other salient aspects of identity.

## Strengths and limitations

A strength of this approach is the breadth of mental health experience represented in the study team. PPI perspectives were sought throughout, including during study design, via an advisory group and in co-design meetings and to assist with dissemination. This facilitated the generation of findings that reflect a variety of perspectives and contexts in which a LoLEM could be hosted. This project also demonstrates the feasibility of a novel and creative approach to mental health programme co-design and theory generation. However, this study does have limitations. Many evidence sources identified in systematic searches were non-mental health specific and involved short-term evaluations. This study principally recruited from the North of England which may limit the transferability of its findings. In a change to the study protocol,[21] theory refinement interviews[23] were not conducted due to limitations in available resources, potentially limiting the explanatory depth of the final programme theory. The reported search on psychological safety was similarly limited to a single multidisciplinary database due to a limited capacity to manage the screening burden of a larger search.

## CONCLUSION

This project is the first to apply realist methods to articulate explanations for how Living Libraries may work and adds to existing literature by identifying specific strategies for promoting the safety and effectiveness of this model as applied to mental health experiences. Continued recognition of the importance of lived experience in shaping mental health policy and practice justifies further consideration of how this approach can be implemented and evaluated.

**Author affiliations**
¹Spectrum Centre for Mental Health Research, Division of Health Research, Lancaster University, Lancaster, UK
²Lancaster University Library, Lancaster University, Lancaster, UK
³Freelance participatory artist, Lancaster, UK
⁴Health Education England North West, Manchester, UK
⁵Patient Experience, Engagement & Safeguarding, Lancashire and South Cumbria NHS Foundation Trust, Preston, UK
⁶Independent survivor researcher, Lancaster, UK
⁷Department of Psychiatry, The University of British Columbia, Vancouver, British Columbia, Canada
⁸Lancaster Medical School, Lancaster University, Lancaster, UK
⁹Faculty of Health and Medicine, Lancaster University, Lancaster, UK
¹⁰Institute of Mental Health, School of Health Sciences, University of Nottingham, Nottingham, UK
¹¹Faculty of Medicine and Health Sciences, Nord University, Namsos, Norway

**Acknowledgements** The research team would like to thank all those who contributed to the co-design of the outputs for this study.

**Contributors** FL designed and led the study, and is the guarantor. PM conducted data collection and analysis for the realist synthesis. JB supported search strategy development. KM and GC co-facilitated experience-based co-design workshops with support from SHJ, RJ, ZG, CL, PJ and LW. JR-M provided methodological guidance for the realist synthesis. SF, CI, HK, SP, EM and MS formed an expert advisory group which provided strategic guidance across the study. SR supported article writing. All authors approved the final article.

**Funding** This work was funded by the National Institute for Health and Care Research (NIHR) under its Research for Patient Benefit (RfPB) Programme (Grant Reference Number NIHR203476). The views expressed are those of the author(s) and not necessarily those of the NIHR or the Department of Health and Social Care. The study was sponsored by Lancashire and South Cumbria NHS Foundation Trust.

**Competing interests** None declared.

**Patient and public involvement** Patients and/or the public were involved in the design, or conduct, or reporting, or dissemination plans of this research. Refer to the Methods section for further details.

**Patient consent for publication** Not applicable.

**Ethics approval** This study involves human participants and was approved by Coventry and Warwick National Health Service Research Ethics Committee (reference number: 305975). Participants gave informed consent to participate in the study before taking part.

**Provenance and peer review** Not commissioned; externally peer reviewed.

**Data availability statement** No data are available.

**ORCID iDs**
Paul Marshall http://orcid.org/0000-0002-3913-3359
Zoe Glossop http://orcid.org/0000-0002-6151-4860
Sarah Powell http://orcid.org/0000-0002-3509-5789

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
