## [Reviewer comments · BMJ Open]

ARTICLE DETAILS

TITLE (PROVISIONAL)	Designing a Library of Lived Experience for Mental Health: integrated realist synthesis and experience-based co-design study in UK mental health services
AUTHORS	Marshall, Paul; Barbrook, John; Collins, Grace; Foster, Sheena; Glossop, Zoe; Inkster, Claire; Jebb, Paul; Johnston, Rose; Jones, Steven; Khan, Hameed; Lodge, Christopher; Machin, Karen; Michalak, Erin; Powell, Sarah; Russell, Samantha; Rycroft-Malone, Jo; Slade, Mike; Whittaker, Lesley; Lobban, Fiona

VERSION 1 – REVIEW

REVIEWER	Long, Janet Australian Institute of Health Innovation, Australian Institute of Health Innovation
REVIEW RETURNED	20-Nov-2023

GENERAL COMMENTS	This is a significant and expertly done realist synthesis and coproduction of an implementation guide for Living Libraries on mental health issues. The concept of Living Libraries is a promising one for reducing stigmatization and social isolation but requires care design and oversight. This study takes a comprehensive approach to providing theory-based, yet operationalized guidance. The authors have followed RAMESES reporting guidelines. The search was comprehensive across black and grey literature. Their initial program theory development was structured around a comprehensive series of interviews and the co-design of the implementation guide was effected using workshops across a 12 month period. Participation was well thought through to include key stakeholders and I applaud the authors on giving this project the generous time frame that was required for robust coproduction. I appreciated the clear and well structured manuscript (carefully proof-read too) and the useful supplementary files. There is not much I can add to this paper – just a couple of discretionary suggestions to consider. Introduction: P.6 Line 31 The concept of a Living Library was new to me and I found the library metaphor both helpful and not helpful. Could you add a few sentences describing the practicalities of the “reader’s” interaction with a “book”? I could get no feel for what is supposed to happen. I read through the references but it was still unclear, probably as there is a lot of variation. Maybe a short example of such an event could clarify. I know more about this is given later but I needed to know it up front in order to understand what you did and to get a feel for the study’s rationale. Figures:
--

	In Figure 1 you have “Wrong language” as an exclusion criterion. Maybe more appropriate to say “Not in English”? References: Ref 33 (Huang et al, 2017) has the Journal name missing. Best wishes for your future research.
--	---

REVIEWER	Sanderson, Kristy University of East Anglia Faculty of Medicine and Health Sciences, Health Sciences
REVIEW RETURNED	23-Nov-2023

GENERAL COMMENTS	This is a very useful contribution to the literature on understanding lived experience-type interventions, with rigorous methods and interest beyond the specific intervention investigated. I have only a couple of minor queries. The initial programme theory interviews were held with advisory group members and PPI lead, all of whom had experience with living libraries. Did any of the workshop participants have such experience? Workshops: the numbers moved around a bit as the workshops progressed. Can the table in supplemental file 5 include how many workshops were attended by each participant? Then a summary sentence could be added to results about extent of contribution for people with lived experience vs carers vs staff, as this may have shaped discussion. Alternatively, in Table 1 the n could be presented in 3 columns for the three participant types so participant mix for each workshop is presented. Limitations: the authors state that project limitations meant that planned theory refinement interviews were not conducted and the search was constrained. Do the authors think these limitations had any substantive impacts on study conclusions and guidance recommendations? Abstract: suggest add "over 12 months" when describing the workshops, to make clear the same group of participants were invited to all 10, rather than 10 concurrent workshops with different participants. Typos: p8 "implantation" - implementation?; p27 "opination" - this is a word but rarely used and presumably not used in a topic guide - opinion?
---

REVIEWER	Milton, Alyssa University of Sydney, Brain and Mind Centre
REVIEW RETURNED	01-Dec-2023

GENERAL COMMENTS	Thank you for the opportunity to review this manuscript. I have never read about living libraries before and find this very interesting with huge potential for building community and service, and individual understanding. This type of co-design work is challenging to synthesise write up, and the authors have done a commendable job synthesising the information in such a clear manner. I have two main suggestions only.
--

	- As I was unfamiliar with the topic of living libraries, I found the abstract objective could be more descriptive. Could the authors please add a definition of what a reader is and what a book is. - I wonder if the author could add reference to a theoretical co-design framework that guide the process of building the model? I understand Gilmore et al. are cited as a guiding framework, but I would like to see a framework more specific to co-design in the methods as well. There is a great deal of literature in this space.
--	---

VERSION 1 – AUTHOR RESPONSE

Reviewer 1

Point 5: The concept of a Living Library was new to me and I found the library metaphor both helpful and not helpful. Could you add a few sentences describing the practicalities of the “reader’s” interaction with a “book”? I could get no feel for what is supposed to happen. I read through the references but it was still unclear, probably as there is a lot of variation. Maybe a short example of such an event could clarify. I know more about this is given later but I needed to know it up front in order to understand what you did and to get a feel for the study’s rationale.

Response: We agree that this concept is likely to be new and potentially unclear for many readers. We have edited the description of a Living Library in the introduction to more clearly articulate what is involved at an event and the roles of Books and Readers, with examples of the kinds of issues discussed at previous events.

Point 6: In Figure 1 you have “Wrong language” as an exclusion criterion. Maybe more appropriate to say “Not in English”?

Response: We have changed this to ‘Not in English’

Point 7: Ref 33 (Huang et al, 2017) has the Journal name missing.

Response: we have added the complete reference

Reviewer 2

Point 8: The initial programme theory interviews were held with advisory group members and PPI lead, all of whom had experience with living libraries. Did any of the workshop participants have such experience?

Response: Yes – we collected information on previous Living Library experience. We have added this in-text at the start of the results section.

Point 9: Workshops: the numbers moved around a bit as the workshops progressed. Can the table in supplemental file 5 include how many workshops were attended by each participant? Then a summary sentence could be added to results about extent of contribution for people with lived experience vs carers vs staff, as this may have shaped discussion. Alternatively, in Table 1 the n could be presented in 3 columns for the three participant types so participant mix for each workshop is presented.

Response: We agree that it may be useful for readers to understand the type of mental health experience represented at each workshop. We have therefore added columns to Table 1 showing the mix of experience (mental health service user, carer, and support roles) represented in each workshop.

Point 10: Limitations: the authors state that project limitations meant that planned theory refinement interviews were not conducted and the search was constrained. Do the authors think these limitations had any substantive impacts on study conclusions and guidance recommendations?

Response: We have added a sentence to the limitations section stating that the explanatory depth of the programme theory may have been limited due to the fact that these interviews were not conducted as planned. However, as these limitations relate primarily to the theory development aspect of the study and given the extensive co-design process, we believe that it is less likely that the final implementation guidance was substantially limited as a result.

Point 11: Abstract: suggest add "over 12 months" when describing the workshops, to make clear the same group of participants were invited to all 10, rather than 10 concurrent workshops with different participants.

Response: 'conducted over 12 months' has been added to the abstract

Point 12: p8 "implantation" - implementation?; p27 "opination" - this is a word but rarely used and presumably not used in a topic guide - opinion?

Response: these words have been corrected to 'implementation' and 'opinion'

Reviewer 3

Point 13: As I was unfamiliar with the topic of living libraries, I found the abstract objective could be more descriptive. Could the authors please add a definition of what a reader is and what a book is.

Response: The first line of the abstract has been extended to provide a clearer definition of the roles of Books and Readers. We have also clarified this in the introduction as per point 5.

Point 14: I wonder if the author could add reference to a theoretical co-design framework that guide the process of building the model? I understand Gilmore et al. are cited as a guiding framework, but I would like to see a framework more specific to co-design in the methods as well. There is a great deal of literature in this space.

Response: We recognise that there are a range of options for selecting co-design models when conducting research of this nature. However, the only model of co-design that guided this study was the Experience-Based Co-Design (EBCD) approach, adapted for integration with realist synthesis to allow theory development to begin with the advisory group interviews. As such, we have re-iterated at the start of the methods section that the study was guided by an integration of the steps used in EBCD and realist synthesis. Regarding the reference to Gilmore et al., this refers to the way in which NVivo was used to organise the initial programme theory framework to support data analysis using realist methods. We have amended the text to clarify this.

VERSION 2 – REVIEW

REVIEWER	Long, Janet Australian Institute of Health Innovation, Australian Institute of Health Innovation
REVIEW RETURNED	06-Jan-2024

GENERAL COMMENTS	Thanks for addressing all my comments. This is a valuable paper.
--

REVIEWER	Milton, Alyssa University of Sydney, Brain and Mind Centre
REVIEW RETURNED	08-Jan-2024

GENERAL COMMENTS	I am happy with the revised version.
--------------------------------------

VERSION 2 – AUTHOR RESPONSE